# Emotional Processing Profile in Patients with First Episode Schizophrenia: The Influence of Neurocognition

**DOI:** 10.3390/jcm11072044

**Published:** 2022-04-06

**Authors:** Verónica Romero-Ferreiro, Lorena García-Fernández, Ana Isabel Aparicio, Isabel Martínez-Gras, Mónica Dompablo, Luis Sánchez-Pastor, David Rentero, Miguel Ángel Alvarez-Mon, Juan Manuel Espejo-Saavedra, Guillermo Lahera, Paloma Marí-Beffa, José Luis Santos, Roberto Rodriguez-Jimenez

**Affiliations:** 1Department of Psychology, Universidad Europea de Madrid, 28670 Madrid, Spain; veronica.romero@universidadeuropea.es; 2CIBERSAM (Biomedical Research Networking Centre in Mental Health), 28029 Madrid, Spain; lorena.garciaf@umh.es (L.G.-F.); aiaparicio@sescam.jccm.es (A.I.A.); monicadompablo@gmail.com (M.D.); davidrente7@hotmail.com (D.R.); guillermo.lahera@gmail.com (G.L.); joseluis.santosg@gmail.com (J.L.S.); 3Instituto de Investigación Sanitaria Hospital 12 de Octubre (Imas12), 28041 Madrid, Spain; isabelmgras@gmail.com (I.M.-G.); lspastor@salud.madrid.org (L.S.-P.); juanmaespejosaavedra@gmail.com (J.M.E.-S.); 4CogPsy Group, Universidad Complutense de Madrid (UCM), 28040 Madrid, Spain; 5Clinical Medicine Department, Universidad Miguel Hernández, 03550 Alicante, Spain; 6Psychiatry Department, Hospital Universitario de San Juan, 03550 Alicante, Spain; 7Psychiatry Department, Hospital Virgen de la Luz, 16002 Cuenca, Spain; 8Neurobiological Research Group, Institute of Technology, Universidad de Castilla-La Mancha, 16071 Cuenca, Spain; 9RETIC (Network of Addictive Conditions), Institute of Health Carlos III, 28029 Madrid, Spain; 10Cardenal Cisneros, Centro de Enseñanza Superior Adscrito a la Universidad Complutense de Madrid, 28040 Madrid, Spain; 11Department of Medicine and Medical Specialities, Faculty of Medicine and Health Sciences, University of Alcalá, 28801 Alcalá de Henares, Spain; maalvarezdemon@icloud.com; 12Department of Psychiatry and Mental Health, Hospital Universitario Infanta Leonor, 28031 Madrid, Spain; 13Ramón y Cajal Institute of Sanitary Research (IRYCIS), 28034 Madrid, Spain; 14Legal Medicine, Psychiatry and Pathology Department, Universidad Complutense de Madrid, 28040 Madrid, Spain; 15Department of Psychiatry, Príncipe de Asturias University Hospital, 28805 Alcalá de Henares, Spain; 16School of Psychology, University of Wales Bangor, Bangor LL57 2AS, UK; pbeffa@bangor.ac.uk

**Keywords:** schizophrenia, first episode, emotional processing, social cognition, MCCB, MSCEIT

## Abstract

This study sought to investigate the influence of neurocognition on the emotional processing profiles of patients with first-episode schizophrenia, using the 4-branch Mayer-Salovey-Caruso Emotional Intelligence Test (MSCEIT) (Perceiving Emotions; Facilitating Emotions; Understanding Emotions and Managing Emotions). A sample of 78 patients with first-episode schizophrenia and a group of 90 non-psychiatric control subjects were included in this work. The initial results showed that patients had lower scores than controls for the “Understanding Emotions” and “Managing Emotions” MSCEIT branches. However, after controlling for neurocognition, the only deficits were found on the “Managing Emotions” branch of the MSCEIT. This branch can be considered as measuring a more sophisticated level of emotional processing, which may constitute a deficit in itself. In conclusion, patients with first-episode schizophrenia present deficits in social cognition at the highest level that seem to be independent from neurocognition. These findings support the inclusion of the “Managing Emotions” branch of the MSCEIT as part of the MCCB.

## 1. Introduction

Social cognition includes theory of mind, social perception, social knowledge, attributional biases, and emotion processing [1]. Some of the most studied aspects of social cognition in schizophrenia are emotion processing and mentalizing. The study of emotion processing analyzes how people perceive and use emotions adaptively in different contexts. Mentalizing refers to the ability to infer the intentions, dispositions, emotions and beliefs of others [2]. People diagnosed with schizophrenia have consistently shown impairments in these aspects of social cognition [3,4], being linked to poor functioning across different stages of the disorder [5,6,7,8]. However, there has been a large debate in the literature about the extent of overlap in schizophrenia between social cognition and other more general aspects of non-social cognition with identifiable neural substrates, commonly referred to as neurocognition [9,10]. Neurocognition refers to cognitive domains that have traditionally been referred to as “cognitive” in the literature, such as speed of processing, working memory, attention, memory, or executive functions [7].There is some evidence supporting the notion that social cognition explains even more variance in community functioning (i.e., interpersonal relations, work functioning) than neurocognition [11], and that it may be a mediator between neurocognition and functional outcome in schizophrenia [1]. In fact, first-episode schizophrenia patients have been found to develop some compensatory strategies for both cognitive and emotional deficits [12,13].

Neurocognition has been extensively studied in schizophrenia [14,15]. The lack of a general consensus regarding the instruments to assess cognitive functioning in this population was one of the reasons that the U.S. National Institute of Mental Health (NIMH) promoted the creation of MATRICS—“Measurement and Treatment Research to Improve Cognition in Schizophrenia” [16]. One of the primary goals of this initiative was to reach a consensus and develop a cognitive battery for use in clinical trials and research. This initiative culminated with the creation of the MATRICS Consensus Cognitive Battery (MCCB) [9,10]. Several studies have been conducted using this battery with first-episode schizophrenia individuals (FESz) [17,18]. The MCCB comprises ten tasks assessing seven cognitive domains that are impaired in schizophrenia. Among them, only one task assesses social cognition (in particular, emotional processing): the Managing Emotions (branch 4) from the Mayer-Salovey-Caruso Emotional Intelligence Test (MSCEIT) [19]. The other three branches of the test (Perceiving Emotions, Facilitating Emotions and Understanding Emotions) were left out.

Several studies have applied the MCCB in FESz, and the results concerning emotional processing are sometimes conflicting. Specifically, some authors have found a relative preservation of these abilities in early-onset schizophrenia compared to controls [17,20], but others have found no differences between first-episode patients and controls [18,21,22,23]. Likewise, some authors found those impairments to be stable across phases of the disorder [24,25], while others found a decline in chronic patients [23], or an even better performance in chronic patients than in FESz [26]. Besides these inconsistent results, there are insufficient studies applying the full version of the MSCEIT to explore whether disturbances in other domains of emotional processing exist in patients with first-episode schizophrenia. The aims of the present study were to obtain a profile of emotional processing in a group of patients with first-episode schizophrenia assessed with the complete MSCEIT, compared with a healthy control group sample, and to study the possible modulatory role that neurocognition could have on social cognition. Specifically, the two hypotheses that were tested were that: (i) FESz patients would show a significant impairment in the four MSCEIT branches compared to controls; and (ii) that those impairments would be modulated by neurocognitive functioning to some extent.

## 2. Materials and Methods

### 2.1. Participants

This cross-sectional study was carried out between 1 March 2020, and 31 December 2021. The sample included 78 FESz outpatients who were consecutively recruited in the First Episode Programs of the Universitary “12 de Octubre” Hospital (Madrid, Spain) and “Virgen de la Luz” Hospital (Cuenca, Spain). A total of 102 patients were initially considered, but 11 refused to participate, 3 were excluded due to poor language comprehension, and 10 for substance use. The inclusion criteria were: (1) diagnosis of schizophrenia or schizophreniform disorder according to DSM-5 criteria [27], using the Structured Clinical Interview for DSM-5 (SCID-5) [28]; (2) at least eight weeks of stabilization on their antipsychotic medication after discharge from the hospitalization unit; (3) age of 18 to 55 years; and (4) sufficient fluency in Spanish to allow them to complete the protocol. Exclusion criteria were: (1) substance abuse/dependence in the past eight weeks (excluding nicotine and caffeine) and using clinical interviews and urine analysis for this purpose; (2) neurological or somatic diseases that could interfere with the performance of the tasks; (3) traumatic head injury; and (4) premorbid IQ score estimated by the Word Accentuation Test (WAT) [29,30] below 70. The clinical sample was compared with 90 healthy control subjects. The inclusion criteria for this group were: (1) age of 18 to 55 years, and (2) sufficient fluency in Spanish to allow them to complete the protocol. Exclusion criteria were the same as for the schizophrenia patients, with the addition of: (5) no diagnosis of any mental disorder according to DSM-5 criteria, and (6) no psychotic disorder as antecedent in first-degree relatives. Controls were selected from cultural associations belonging to the same geographical area as the patient group. Both patients and controls were clinically assessed by experienced researchers who have used the scales for more than 5 years. The study was approved by the Clinical Research Ethics Committee of the Hospital 12 de Octubre, and all participants signed an informed consent form. The demographics and clinical characteristics of the patients and healthy controls are presented in Table 1.

### 2.2. Instruments

Symptoms were assessed for descriptive purposes using the Positive and Negative Syndrome Scale (PANSS) [31]. Emotional processing was evaluated using the Mayer-Salovey-Caruso Emotional Intelligence Test (MSCEIT) which consists of 8 subscales assessing 4 components (branches) of emotion processing [19]. The first branch, Perceiving Emotions, has 2 subscales measuring emotion perception in faces and pictures (e.g., identifying the degree to which certain feelings are expressed by a color photograph of a human face). The second branch, Facilitating Emotions, is derived from 2 subscales examining how mood enhances thinking and reasoning, and which emotions are associated with which sensations (e.g., asking subjects to evaluate the usefulness of different emotions that would best assist a specific cognitive task and behavior). The third branch, Understanding Emotions, has 2 subscales that measure the ability to comprehend emotional information, including blends and changes between and among emotions (e.g., asking participants to select which 1 of 5 emotions best describes a situation). The fourth branch, Managing Emotions, has 2 subscales that examine the regulation of emotions in oneself and in relationships with others by presenting vignettes of various situations, along with ways to cope with the emotions depicted in these vignettes. For the current study, we examined the 4 MSCEIT branch scores corrected for age and gender. Finally, cognitive performance was evaluated using the MCCB which assesses seven cognitive domains: Speed of Processing, Attention/Vigilance, Working Memory, Verbal Learning, Visual Learning, Reasoning and Problem Solving, and Social Cognition [9,10]. This battery allows the neurocognition score to be calculated by combining these different neurocognitive domains, excluding the social cognition domain. This study used the published and approved translation of the MCCB for Spain and the Spanish normative and standardized data correction [32]. With the objective of controlling the possible effect of neurocognition, the MCCB Neurocognition T-score, including all domains except social cognition, was also calculated for each participant. Age and gender correction for normative scoring were used, following the recommendations by the co-norming and standardization guidelines [9].

### 2.3. Statistical Analysis

Data were managed and analyzed with SPSS v.24. Raw data from each branch of MSCEIT were corrected according to age and gender. Similarly, raw scores from each test of MCCB were entered into the MCCB Computer Scoring Program to produce age- and gender-corrected T-scores. These data were submitted to a two (group: patients, controls) by four (MSCEIT branches) mixed model analysis, with random intercept for each subject and an identity covariance structure. The group x branch interaction was analyzed with an estimated marginal means post hoc analysis with the Bonferroni adjustment. As MSCEIT and MCCB scores were standardized according to age and gender, it was not felt necessary to include them as covariates in the analysis, even though differences between groups were found. However, years of education was included in the models, as there were differences between groups, and they were not controlled by standardized scores.

Finally, the same analysis was repeated including the MCCB Neurocognition T-score as a covariate, with the aim of studying the influence of neurocognition on emotional processing. Collinearity diagnostics were based on the variance inflation factor (VIF). Given that all VIF values were lower than 1.4, we can assume that there were no effects of collinearity.

## 3. Results

### 3.1. Descriptive Statistics

As can be seen in Table 1, there were no differences between patients and controls in terms of age *t*(166)= 1.56, *p* = 0.12. The distribution of gender differed across groups χ^2^ = 8.89, *p* = 0.003. Patients and controls also differed in years of education *t*(159) = 4.87, *p* < 0.001. Mean age- and gender-corrected T-scores for each MSCEIT branch and MCCB domains and neurocognition scores are presented in Table 2.

### 3.2. Mixed Model Analysis

#### 3.2.1. Comparison between FESz and Controls in the Four Branches of the MSCEIT

There was a significant effect for the MSCEIT branch (*F*(3, 476) = 26.51, *p* < 0.001), and a significance for years of education (*F*(1, 158) = 15.92, *p* < 0.001). No significance of group was found (*F*(1, 158) = 2.69, *p* = 0.103). A significant interaction between the groups and the MSCEIT branch was found (*F*(3, 476) = 7.85, *p* < 0.001). Pairwise comparisons between groups showed that FESz patients had lower scores than the control group in Understanding Emotions (*p* = 0.011, mean diff = −5.97, 95%CI: −10.57; −1.37) and in Managing Emotions (*p* < 0.001, mean diff = −8.56, 95%CI: −13.17; −3.95). Neither Perceiving Emotions nor Facilitating Emotions showed differences between FESz and controls (*p* = 0.409 and *p* = 0.449, respectively).

#### 3.2.2. Comparison between FESz and Controls in the Four Branches of the MSCEIT Controlling for Neurocognition Effects

After including the MCCB neurocognitive score as a covariate, the results were as follows: the main effect of MSCEIT branch (*F*(3, 476) = 26.48, *p* < 0.001), and years of education remained significant (*F*(1, 157) = 7.50, *p* = 0.007). Neurocognition revealed a significant effect on the model (*F*(1, 157) = 5.32, *p* = 0.022). Again, there was no main effect of group (*F*(1, 157) = 0.007, *p* = 0.933). Finally, the group by MSCEIT branch interaction was significant (*F*(3, 494) = 8.74, *p* < 0.001). Pairwise comparisons between groups showed that FESz patients had lower scores than the control group only in Managing Emotions (branch 4) (*p* = 0.02, mean diff = −6.00, 95%CI: −11.08; −0.93). There were no differences between patients and controls in Perceiving Emotions, Facilitating Emotions and Understanding Emotions (*p* = 0.083, *p* = 0.095 and *p* = 0.182, respectively).

## 4. Discussion

The main objective of this study was to investigate emotional processing of first-episode schizophrenia patients, considering the possible modulating effects of neurocognition.

Initially, results showed deficits in the schizophrenia patient group compared to controls in the branches measuring Understanding and Managing Emotions, which represent emotional abilities that require higher level cognitive processing. Importantly, when the MCCB neurocognition scores were included in the analysis, the effect changed, and the deficits were only observed in the Managing Emotions branch. This result implies that the poor capacity to understand emotional information showed by first-episode schizophrenia patients was accounted for, partially, by neurocognitive deficits. When the neurocognitive performance was controlled for, only the differences in the ability to regulate emotions with themselves and others remained significant. Our results show that emotional regulation entails a relatively independent deficit in first-episode schizophrenia, and is not linked to other aspects of general cognitive deficits. This result suggests that higher-order social cognition abilities might be controlled and regulated by a specific neural circuit. This finding also supports the inclusion of this single branch of the MSCEIT as part of the MCCB.

Literature comparing emotional processing between first-episode patients and controls with MSCEIT have mostly used the Managing Emotions branch as a sole measure of emotional processing, generally revealing deficits in the patients’ group [22]. Some other authors, however, did not found this pattern, which could be explained by the differences in our experimental design. For example, some used relatively smaller sample sizes (*n* = 31 patients and *n* = 67 controls) [20];], or included patients with schizoaffective disorder, depressed type [17], making a close comparison difficult. As far as we know, only one study has compared patients and controls using the four branches of the MSCEIT. This study assessed a sample of three phases of psychotic illness: prodromal, first episode, and chronic schizophrenia, and a control group. They found deficits between patients (as a whole) and controls in the four branches, but none of them changed across the phases [24]. It is also important they did not study the potential effect of neurocognition. This may be the reason why their results do not correspond with those reported in the present study.

Finally, apart from understanding emotions, patients with first-episode schizophrenia in our study demonstrated preserved abilities regarding emotion perception (Perceiving Emotions) in line with previous research [33], and an appropriate evaluation of how different emotions guide behavior (Facilitating Emotions). These results need to be taken into account when designing cognitive remediation programs to improve social cognition in schizophrenia. Hypothetically, these programs should focus on (apart from, of course, neurocognition) the management of emotions, that is, in the “strategic” sector of emotional intelligence which entails high-level thought processing [34]. Patients with FESz could better benefit from broad social cognition training with a pragmatic, ecological, and action-based approach.

The main strengths of this study lay in the relatively large sample size, the use of the complete MSCEIT, and the use of MCCB to control the effect of neurocognition (and years of education) on the emotional processing scores. Despite this, there were some differences between the groups in terms of gender and years of education; however, their impact on our results was negligeable, as both MSCEIT and MCCB scores were corrected for age and gender. In fact, we performed an analysis including age and gender as covariates, but this had no impact on the results (data not shown). Additionally, given that some clinical manifestations of schizophrenia are earlier in men than women, and thus, available evidence suggest that their neurocognitive and social cognition abilities may differ, we reconducted the analyses using only the male participants of our total sample. The results were essentially the same as those exposed in this study, with variations according to the loss of statistical power. However, this should be considered a preliminary result. Future studies will be needed to address gender-specific differences in the social cognition of FESz patients. The results reported here using MSCEIT are important for evaluating the emotional processes in first-episode schizophrenia patients, but further studies should include other aspects of social cognition, such as social perception, theory of mind, or attributional bias [35].

## 5. Conclusions

This study shows that first-episode schizophrenia patients are selectively impaired in emotional processing when this requires high-level cognition, paired with other more general deficits in neurocognition. Current evidence supports the specific inclusion of the Managing Emotions branch when using MCCB, and the assessment of neurocognition in experimental and clinical settings.

## Figures and Tables

**Table 1 jcm-11-02044-t001:** Mean (SD) of demographic and clinical characteristics of participants.

	FESz (*n* = 78)	Controls (*n* = 90)	Statistics
Age years	26.23 (7.3)	27.97 (7.0)	*t* = 1.56, *p* = 0.12
Sex, *n* (% male)	55 (70.5%)	43 (47.8%)	χ^2^ = 8.89, *p* = 0.003
Education years	12.0 (3.0)	14.2 (2.9)	*t* = 4.87, *p* < 0.001
PANSS—Positive	10.6 (4.5)		
PANSS—Negative	16.7 (8.0)		
PANSS—General Psychopathology	27.9 (8.7)		
CPZ ^1^	403.9 (246.8)		
Duration of untreated psychosis (days)	168.9 (186.7)		

^1^ Chlorpromazine equivalent dose (mg/day).

**Table 2 jcm-11-02044-t002:** Mean (standard deviations) of T-scores of the MSCEIT and MCCB domains (excluding social cognition) and Neurocognition. FESz = first episode of schizophrenia.

	FESz (*n* = 78)	Controls (*n* = 90)	*t* Test (*p* Value)
MSCEIT Perceiving Emotions	104.97 (14.3)	105.18 (14.4)	*t*(166) = 0.927 (0.927)
MSCEIT Facilitating Emotions	101.35 (14.3)	102.02 (14.9)	*t*(166) = 0.298 (0.766)
MSCEIT Understanding Emotions	90.32 (14.4)	98.8 (14.1)	*t*(166) = 3.85 (<0.001)
MSCEIT Managing Emotions	91.12 (14.1)	102.31 (15.7)	*t*(165) = 4.81 (<0.001)
MCCB Speed of Processing	35.6 (9.0)	51.9 (8.8)	*t*(165) = 11.75 (<0.001)
MCCB Attention/Vigilance	34.2 (8.8)	48.2 (9.5)	*t*(165) = 9.83 (<0.001)
MCCB Working Memory	38.1 (10.3)	49.3 (10.8)	*t*(165) = 6.85 (<0.001)
MCCB Verbal Learning	31.0 (13.9)	45.0 (11.8)	*t*(165) = 7.07 (<0.001)
MCCB Visual Learning	34.6 (15.2)	47.0 (10.8)	*t*(137) = 5.97 (<0.001)
MCCB Reasoning and Probl. Solving	40.8 (10.6)	52.6 (7.7)	*t*(139) = 8.16 (<0.001)
MCCB Neurocognition	30.54 (11.8)	48.27 (10.2)	*t*(153) = 10.30 (<0.001)

## Data Availability

The data from this study will be made publicly available following the completion of all publications and following the removal of all identifiers by request.

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
