# Peer review of "Emotional Processing Profile in Patients with First Episode Schizophrenia: The Influence of Neurocognition"

_jcm, 2022, doi:10.3390/jcm11072044_

Round 1

Reviewer 1 Report

Although this is an interesting paper, there are some serious methodological issues that diminish the quality of the paper. First, gender and education are different in the control group and the patient group. This is a problem, as controls' cognition/neuropsychological performance is widely known that is affected by these two variables, so no comparisons can be made. An alternative could be to select controls and patients with simlar means in the demographics and rerun the statistical analyses. or the researchers should rerun the testing by selecting controls with similar demographics. Second, the neuropsychological assessment should be more detailed. MCCB is useful, but additional standardized tests should be added. This study seems more as a preliminary report. Third, the recruitment procedure is not described for both groups of participants. Fourth, the age range is huge, and conclusions should also take into consideration analyses with the age parameter in them.

Author Response

Reviewer 1

Although this is an interesting paper, there are some serious methodological issues that diminish the quality of the paper. First, gender and education are different in the control group and the patient group. This is a problem, as controls' cognition/neuropsychological performance is widely known that is affected by these two variables, so no comparisons can be made. An alternative could be to select controls and patients with similar means in the demographics and rerun the statistical analyses. or the researchers should rerun the testing by selecting controls with similar demographics.

We totally agree Reviewer 1, both gender and years of education affect cognitive performance. For that reason, we included normative age-and gender-corrected T-scores rather than raw scores. T-scores are standard scores with M=50 and SD=10 that precisely make results from different groups more comparable. We have emphasised this aspect in the revised version of the manuscript (Methods section (instruments), paragraph 1).

Concerning the educational level, the MATRICS initiative recommend correcting scores for age and gender, but not for educational level, given that it is not only a source of noise as the disorder itself influences the academic level achieved (Kern et al., 2008a; see the MCCB Scoring Program: consideration of age, gender, and education effects section). Regardless of this, motivated by intellectual curiosity we have selected a subset of individuals (64 controls and 70 patients) equalled in terms of gender (27 females vs 21 females, respectively) and years of education (M=13.2, SD=2.4 vs. M=12.4 SD=2.7, respectively) and the outcome of the regression analysis did not change.

Kern, R. S., Nuechterlein, K. H., Green, M. F., Baade, L. E., Fenton, W. S., Gold, J. M., ... & Marder, S. R. (2008a). The MATRICS Consensus Cognitive Battery, part 2: co-norming and standardization. American Journal of Psychiatry165(2), 214-220.

Second, the neuropsychological assessment should be more detailed. MCCB is useful, but additional standardized tests should be added. This study seems more as a preliminary report.

Of course, other neuropsychological measures instead of the MCCB could have been used. We chose the MCCB, because it is a consensus battery widely used worldwide, developed by the MATRICS initiative of the NIH of the United States (Kern et al., 2008b). The process of standardization and obtaining normative data has been carried out in the Spanish population by our own group (Rodriguez-Jimenez et al., 2012). This battery produces a Neurocognition score by combining different neurocognitive domains, which aligns directly to the objective of our work. In the revised manuscript, we point out this aspect in the section on methodology: instruments (Methods section (instruments), paragraph 1).

Nuechterlein, K. H., Green, M. F., Kern, R. S., Baade, L. E., Barch, D. M., Cohen, J. D., ... & Marder, S. R. (2008b). The MATRICS Consensus Cognitive Battery, part 1: test selection, reliability, and validity. American Journal of Psychiatry165(2), 203-213.

Rodriguez-Jimenez, R., Bagney, A., Garcia-Navarro, C., Aparicio, A. I., Lopez-Anton, R., Moreno-Ortega, M., ... & Palomo, T. (2012). The MATRICS consensus cognitive battery (MCCB): co-norming and standardization in Spain. Schizophrenia research134(2-3), 279-284.

Third, the recruitment procedure is not described for both groups of participants.

We highly appreciate this suggestion. The revised manuscript describes in more detail the process in both groups (Methods section (participants), paragraph 1).

Fourth, the age range is huge, and conclusions should also take into consideration analyses with the age parameter in them.

It is true that the age range is huge. We have set the upper limit at 55 years, although as can be inferred from the average age of both groups, they are essentially composed by young participants. As previously noted, there are no statistical differences between the two groups in terms of age, but also, we have corrected MSCEIT and MCCB raw scores by gender and age. Consequently, the possible effect of age on results would be minimal. In the revised manuscript we explain this point (Methods section (statistical analysis), paragraph 1).

Finally, it should be added that an extensive language review has been made by one bilingual co-author. 

Reviewer 2 Report

The authors cleverly employed a neat experimental paradigm and a system of regression models to investigate the difference in emotional processing abilities between healthy controls and first-episode schizophrenic patients while accounting for their respective neurocognitive structures. The authors' aims are clearly stated and are justified by the literature.

Relevant considerations on the Theory

Some parts of the manuscript seem contracted in the sense that they are highly narrowed, synthetised. I recognize that minimizing space and avoiding excessive wording are noble intents. However, the authors should better introduce the concepts they use and the general flow of the manuscript.

Please briefly explain the term "Neurocognition" when you first employ it. Throughout the text, the authors should also explain which specific type of neurocognitive function is treated in more detail.

Moreover, community functioning is an essential and multifaceted construct. The manuscript will be more appealing to the readers if the authors introduce and discuss in the appropriate sections which aspect(s) of community functioning are involved.

Relevant considerations on the Methods

Although most subsections of the Methods are sufficiently clear, the authors did not mention how they preprocessed their data.

- Were dependent/independent variables handled before linear regression models?

- Did the authors check the collinearity statistics?

- There is consensus that multiple linear regressions require covariate centering. Did the author address this?

- Please report any data preprocessing, perform analyses using covariate centering, and check for collinearity issues. If the authors obtain different results in these models, they may want to conclude based on both the procedures (with and without covariate centering).

Author Response

Reviewer 2

Relevant considerations on the Theory

Some parts of the manuscript seem contracted in the sense that they are highly narrowed, synthetised. I recognize that minimizing space and avoiding excessive wording are noble intents. However, the authors should better introduce the concepts they use and the general flow of the manuscript. Please briefly explain the term "Neurocognition" when you first employ it. Throughout the text, the authors should also explain which specific type of neurocognitive function is treated in more detail. Moreover, community functioning is an essential and multifaceted construct. The manuscript will be more appealing to the readers if the authors introduce and discuss in the appropriate sections which aspect(s) of community functioning are involved.

As the reviewer rightly points out, we tried to give the most information using the minimum space. Following his/her suggestion, in the revised manuscript we have expanded some definitions such as neurocognition, social cognition in the introduction section (Nuechterlein et al., 2004; González-Ortega et al., 2020). Also, in the instruments section, we have clarified that neurocognition includes Speed of Processing, Attention/Vigilance, Working Memory, Verbal Learning, Visual Learning, Reasoning and Problem Solving, that is: all domains of the MCCB except social cognition (Methods section (instruments), paragraph 1). Finally, community functioning is very important in the treatment of patients and their quality of life, especially in those with a first episode. It is a multifaceted construct that includes different aspects. Even though we have not worked with that variable, given its importance in the introduction section, we have pointed out its important relationship with social cognition, as well as its possible mediating role between neurocognition and functional outcome in schizophrenia (Introduction section, paragraph 1).

González-Ortega, I., González-Pinto, A., Alberich, S., Echeburúa, E., Bernardo, M., Cabrera, B., ... & Selva, G. (2020). Influence of social cognition as a mediator between cognitive reserve and psychosocial functioning in patients with first episode psychosis. Psychological Medicine50(16), 2702-2710.

Nuechterlein, K. H., Barch, D. M., Gold, J. M., Goldberg, T. E., Green, M. F., & Heaton, R. K. (2004). Identification of separable cognitive factors in schizophrenia. Schizophrenia research72(1), 29-39.

Relevant considerations on the Methods

Although most subsections of the Methods are sufficiently clear, the authors did not mention how they pre-processed their data.

- Were dependent/independent variables handled before linear regression models?

In response to this concern raised by the reviewer, the only pre-processing data carried out was to eliminate some participants (patient screening has been included in the methods section of the revised manuscript (Methods section (participants), paragraph 1) and correcting MSCEIT and MCCB raw scores by age and gender, which has been also included in the manuscript (Methods section (instruments), paragraph 1).

- Did the authors check the collinearity statistics?

We thank Reviewer 2 for pointing out this omission. We checked the collinearity Yes, it has been now included in the new version of the manuscript (Methods section (statistical analysis), paragraph 2).

- There is consensus that multiple linear regressions require covariate centering. Did the author address this?

We thank Reviewer 2 for this valuable suggestion. When planning the statistical approach, we considered to perform covariate centering, which, as the reviewer points out, some published works recommend it (Bell et al., 2018). However, the existence of other authors that indicate some controversy in this regard (Iacobucci et al., 2016; Wickens & Keppel, 2004) lead us to consult with experts to know their opinion. They pointed out that, although mean centering has several advantages (such as resolve certain multicollinearity issues), when the distribution of a covariate (for instance, neurocognition) is substantially different across the groups of interest (in this case, patients vs. controls), this variable is likely to be highly confounded (i.e., correlated) with the grouping variable itself. Thus, difference in MSCEIT scores between groups, if significant, might be partially (or even totally) attributed to the effect of neurocognition. If the neurocognition effect is modelled by centering around each group’s respective mean it would be difficult to interpret the results: the group difference may be compounded with the effect of neurocognition difference across the groups. Finally, the existence of published works, with an analogous approach to ours, that have not included covariate centering, added to the fact that we do not have collinearity issues, led us to not to perform this procedure.

Bell, A., Jones, K., & Fairbrother, M. (2018). Understanding and misunderstanding group mean centering: A commentary on Kelley et al.’s dangerous practice. Quality & Quantity52(5), 2031-2036.

Iacobucci, D., Schneider, M. J., Popovich, D. L., & Bakamitsos, G. A. (2016). Mean centering helps alleviate “micro” but not “macro” multicollinearity. Behavior research methods48(4), 1308-1317.

Wickens, T. D., & Keppel, G. (2004). Design and analysis: A researcher's handbook. Upper Saddle River, NJ: Pearson Prentice-Hall.

- Please report any data preprocessing, perform analyses using covariate centering, and check for collinearity issues. If the authors obtain different results in these models, they may want to conclude based on both the procedures (with and without covariate centering).

All the methodological issues suggested by Reviewer 2 have been included or discussed, and they have been included when appropriate in the new version of the manuscript.

Reviewer 3 Report

In the present study the Authors aimed to obtain a profile in emotional processing assessed with the complete Mayer-Salovey-Caruso Emotional Intelligence Test (MSCEIT) in a group of patients with first episode of schizophrenia compared with a healthy control group sample; and to study the possible modulatory role that Neurocognition could have in social cognition. Specifically, the Authors tested two hypotheses (1) if persons with first episode of schizophrenia (FESz) would show a significant impairment in the 4 MSCEIT branches compared controls and (2) if that those impairments would be modulated by neurocognitive functioning to some extent.

Overall, I found this study timely, original, well conducted and scientifically sound. I have some suggestions aimed to improve the quality of the paper and these are outlined below:

1) In the introduction a brief note on the cognitive impairment and emotional disturbances associated with first-episode schizophrenia and their repercussions on everyday clinical practice should be added with appropriate references (see dois: 10.1016/j.neuropsychologia.2012.02.005 and 10.1159/000366133).

2) The present cross-sectional study was carried out on 78 FESz outpatients, who were consecutively recruited in the First Episode Programs of the Universitary “12 de Octubre” Hospital (Madrid, Spain) and “Virgen de la Luz” Hospital (Cuenca, Spain). But, how many subjects were screened, but refused to participate? As well, how many FESz outpatients were abusing drugs and thus excluded.

3) Please, add the time frame of the study.

3) Concerning exclusion criterion 1 (substance abuse/dependence in the past eight weeks), how this was assessed? Objectively or simply through patients' reports? And how many subjects were excluded considering this criterion? I believe that many FESz are due to current or past drug use/misuse, so this is an interesting information.

4) Besides, was the presence of an intellectual disability assessed and how? 

5) I believe that Table 1 should contain more clinical informations on participants at baseline (as illness duration, DUP, previous treatments, previous suicide attempts if any and so on).

Author Response

Reviewer 3

In the present study the Authors aimed to obtain a profile in emotional processing assessed with the complete Mayer-Salovey-Caruso Emotional Intelligence Test (MSCEIT) in a group of patients with first episode of schizophrenia compared with a healthy control group sample; and to study the possible modulatory role that Neurocognition could have in social cognition. Specifically, the Authors tested two hypotheses (1) if persons with first episode of schizophrenia (FESz) would show a significant impairment in the 4 MSCEIT branches compared controls and (2) if that those impairments would be modulated by neurocognitive functioning to some extent.

Overall, I found this study timely, original, well conducted and scientifically sound. I have some suggestions aimed to improve the quality of the paper and these are outlined below:

1) In the introduction a brief note on the cognitive impairment and emotional disturbances associated with first-episode schizophrenia and their repercussions on everyday clinical practice should be added with appropriate references (see dois: 10.1016/j.neuropsychologia.2012.02.005 and 10.1159/000366133).

We thank Reviewer 3 for the suggestion. We have included a mention about cognitive impairment and emotional disturbances in FES and the respective references (Introduction section, end of 1st paragraph).

2) The present cross-sectional study was carried out on 78 FESz outpatients, who were consecutively recruited in the First Episode Programs of the Universitary “12 de Octubre” Hospital (Madrid, Spain) and “Virgen de la Luz” Hospital (Cuenca, Spain). But how many subjects were screened, but refused to participate? As well, how many FESz outpatients were abusing drugs and thus excluded.

A total of 102 patients were initially selected. From them, 11 refused to participate, three were excluded due to poor language comprehension, and 10 for substance use. It should be borne in mind that patients were in First Psychotic Episodes program, where one of the main objectives was to avoid the consumption of toxics. In the revised manuscript the screened participants are indicated (Methods section (participants), paragraph 1)

3) Please, add the time frame of the study. 

The dates have been included in the manuscript (Methods section (participants), paragraph 1).

3) Concerning exclusion criterion 1 (substance abuse/dependence in the past eight weeks), how this was assessed? Objectively or simply through patients' reports? And how many subjects were excluded considering this criterion? I believe that many FESz are due to current or past drug use/misuse, so this is an interesting information.

We can do no other than agree with Reviewer 3 at this point. A total of 10 patients were excluded for drug consumption. This was assessed by asking patients and the relatives but also using urine tests. It has been included in the new version of the manuscript (Methods section (participants), paragraph 1).

4) Besides, was the presence of an intellectual disability assessed and how?

We used the Word Accentuation Test validated in Spain (Gomar et al., 2011). A score below 70 was an exclusion criterion. Neither patients nor controls had a score below 70. This exclusion criterion is included in the revised manuscript (Methods section (participants), paragraph 1).

Gomar, J. J., Ortiz-Gil, J., McKenna, P. J., Salvador, R., Sans-Sansa, B., Sarró, S., ... & Pomarol-Clotet, E. (2011). Validation of the Word Accentuation Test (TAP) as a means of estimating premorbid IQ in Spanish speakers. Schizophrenia Research, 128(1-3), 175-176.

5) I believe that Table 1 should contain more clinical information on participants at baseline (as illness duration, DUP, previous treatments, previous suicide attempts if any and so on)

We have included the duration of the disease and DUP. There were no previous treatments because patients are FESz (Table 1).